# Study of Energy Transition Paths and the Impact of Carbon Emissions under the Dual Carbon Target

**Kun Wang [1], Li Ouyang [2],\*** and **Yue Wang [3]**

1    School of Economics and Finance, Xi'an Jiaotong University, Xi'an 710061, China
2    School of Literature and Media, Jingchu University of Technology, Jingmen 448000, China
3    Normal College, Jingchu University of Technology, Jingmen 448000, China
\*    Correspondence: yangmin@jcut.edu.cn

**Abstract:** In recent years, the world's environmental problems have become increasingly serious, and energy transition and carbon emission issues have gradually gained attention from various countries. China has promulgated several policies and adopted various reform measures to achieve a comprehensive energy transition and carbon neutrality as soon as possible. Therefore, this study makes researches and forecasts the energy transition and carbon emissions in China under the dual carbon target. A LEAP (Long range Energy Alternatives Planning) model is developed to analyze the energy parameters of Beijing under various scenarios and to provide a quantitative analysis basis for the energy transition path. The obtained experimental results indicate that the energy demand under the partial energy transition scenario and full energy transition scenarios are 68.651 million tons of standard coal and 75.759 million tons of standard coal, respectively, forming an effective control, while the carbon emissions both achieve the carbon peak in 2025 at 81.903 million tons and 80.624 million tons respectively, and achieve 46.588 million tons in 2060. The carbon-neutral pathway reaches the carbon peak in 2025, and approaches full energy transition in 2035, and finally reaches the full energy transition control effect in 2060. To date, most of the relevant studies have been conducted in a qualitative way, and the amount of quantitative analysis is insufficient. At the same time, research on the development path specifically at the city level is relatively insufficient as well. Therefore, the present study can provide a theoretical basis for specifying the promotion method of urban energy transformation and the path of carbon emission reduction.

**Keywords:** energy transition; carbon reduction; long-term energy alternative planning system; scenario design





## 1. Introduction

With the connection of global social demand markets and the deepening of global industrial development and transfer, the problems of climate change and environmental pollution are becoming increasingly serious. The old consumption mode of non-renewable resources causes excessive consumption of limited resources and serious environmental pollution, which is an unsustainable development mode. Therefore, energy transformation, as a strategy that can achieve benign improvement in both environmental pollution and resources, is currently being constantly tried [1–3]. As an important development goal and feasible path for energy transformation, energy conservation, and emission reduction in the future, adual-carbon target requires achieving both breakthrough development in energy production technology and fundamental change in the energy consumption structure, that is, achieving energy conservation and emission reduction from both energy production and energy consumption. At present, research on the dual-carbon target has been gradually deepening [4–6].

At present, the research on energy transformation is mainly divided into two aspects; one is to explore from the technical perspective of energy cleaning, the other is to explore

from the perspective of the social and industrial structure. From the perspective of clean energy technology, Cheng Z. takes the dual-carbon goal as the strategic goal and focuses on the separation and clean and efficient utilization of low-rank coal. Low-rank coal can show strong compatibility and adsorption capacity through effective surface agent improvement, and can provide an auxiliary path for the realization of the dual-carbon goal [4]. In the context of the dual-carbon goal, the team of Zou C. carried out research from the perspective of sustainable development of mining enterprises, and reduced the economic cost and energy consumption caused by mineral processing and grinding using ceramic ball replacement technology in anore grinding machine to achieve the dual improvement of economic and industrial benefits while ensuring control of the grinding fineness [5]. The team of Wen W. started with electricity, the only secondary clean energy, and took the dual-carbon goal as their strategic goal to build the main path of highly clean energy internet. Through use of the internet and the Internet of Things for real-time adjustment of power supply, they were able to achieve coordinated control of the power grid, laying the foundation for the stable development of the electrified clean energy network [7]. It can be seen that although the development of the detailed technology of energy cleaning is essential, the support of the detailed technology can only play a supporting role at the bottom level in the regional energy transformation. In the regional energy transformation, especially the urban energy transformation, the guidance of policy means is essential, otherwise the area that technology can affect is always limited. The policy is closely related to the energy industry and the energy consumption market. The Idris M N M team took the utilization of bioenergy in line with the characteristics of the local energy structure in Malaysia as an example, explored the possibility of renewable energy sector development policies, analyzed the stability of the energy and bioenergy market under different financial mechanisms, combined the bioenergy policy with the local carbon dioxide emissions, and analyzed the positive role of energy adjustment policy support for carbon emissions [8]. Jiang T. analyzed the structural emission reduction between China's power industry and thermal industry from the perspective of energy input and output under the dual-carbon target, taking the energy utilization method and the input–output method as the main decomposition structure of the study. Their research results show that the optimized energy intensity, energy composition, and energy input mode are conducive to minimizing the carbon dioxide emissions of China's current heating and power sectors. They then used this minimization scheme to ensure energy conservation and emission reduction effect under the transition phase of the dual-carbon target [9]. This study selected Beijing as the main object. Similar to Jiang T., they explored the emission reduction of the thermal and power sectors. It was not limited to the thermal and power sectors, including the transportation sector, industrial sector and living sector in the analysis and prediction framework. At the same time, with the same idea as the Idris M N M team, their research created a more realizable energy adjustment scenario based on the local natural renewable energy structure characteristics of the background, and analyzed it under this scenario. Only the analysis based on the characteristics of the local energy structure in the study area and the characteristics of local energy consumption is the integrity analysis of the development possibility of the energy market. Based on this point, this study takes both into account.

On the basis of energy transformation, the research mainly analyzes carbon emissions. Carbon emissions have always been an important environmental issue accompanying economic development, and green economy is the inevitable path of global economic development. Li Z Z et al. analyzed the distribution of green investment in 30 provinces in China from 1995 to 2017 from a macro perspective, and tried to determine the role of key macroeconomic factors in reducing carbon emissions. In the process of research, the long-term correlation analysis between variables was realized through the co-integration empirical method. The research results show that the effective layout of green investment projects can achieve long-term and short-term carbon emissions reduction, and that there is a two-way causal relationship between the two [10]. At the same time, the team of Sun L. studied the impact of China's domestic trade on the transfer of carbon emissions between different provinces. Based on the social network analysis model and the multi-regional

input-output model, their study explored the characteristics of carbon emissions transfer in different departments under different provincial environments. The research results showed that the eastern region has a more obvious two-way spillover effect, the western region plays an intermediary role, and the central region mainly plays an inflow role [11]. It can be seen that when only discussing carbon emissions and regional space issues, the issue of how to limit carbon emissions cannot be implemented and analyzed. This is because carbon emissions are not only a single pollution issue; they are closely related to the local energy industry, energy consumption structure, and local environment and energy policies. Most global carbon emissions come from the consumption of non-renewable energy, and as such it is necessary to adjust the regional energy structure. In this field, Zheng J. et al. have analyzed the change of carbon dioxide emissions caused by the upgrading of domestic industrial structure and energy efficiency by exploring China's regional economic and energy development and change model under the dual-carbon goal. Their research results show that the industrial structure and energy composition are indeed important driving factors for the reduction of carbon emissions at the national level, and that the stabilization of carbon emissions can be effectively maintained through interregional cooperation [12]. Yuping L et al. took Argentina as their main research object and analyzed the dynamic image of renewable and non-renewable energy consumption on Argentina's carbon emission level in 1970 and 2018. The research results show that the coordinated development of renewable energy consumption and global economy can effectively reduce carbon emissions [13]. The team of Sharif A. analyzed changes in carbon emissions under a combination of renewable energy and non-renewable energy consumption using heterogeneous panel causality analysis. The research results show that all variables present a long-term comprehensive nature. At the same time, the degree of degradation of financial development is conducive to lower carbon emissions, while the positive development of finance has a negative impact on carbon emissions [14]. It can be seen that there is a linkage triangle relationship among the three elements of economic development, energy consumption, and carbon emissions. To reduce carbon emissions while developing the economy, it is necessary to adjust the energy consumption and production market as a whole, change the old energy path dependence, and improve the energy structure. The research of Sharif A and Zheng J confirms this point. The present study is based on the correlation between energy structure adjustment and carbon emissions, taking Beijing as the main research area, the development trend of energy economy as the main scenario hypothesis, and analyzing the impact of the energy transformation path on carbon emissions by building a long-term energy substitution planning system to achieve regionalization analysis and regionalization development prospect prediction. In this way, we provide a theoretical basis for transformation planning for similar cities under the global energy transformation environment.

## 2. Materials and Methods

### 2.1. LEAP Model Scenario Building

For the main path of energy transformation in large cities, Beijing was selected as the main research object. This is because the energy transformation characteristics of "coal removal, gas extraction and power generation" formed in the process of renewable energy development in Beijing are in line with the scientific energy transformation trend of taking natural gas as the transitional energy and then moving to the road of electrification. The current energy transformation trend of Beijing can provide guidance for large and medium-sized cities to transform from coal resource dependence to a new energy transformation strategy of energy dependence. On this basis, the LEAP model is selected as the main application model. The LEAP model, or Long Term Energy Alternatives Planning System, is a vertical end-use energy consumption model based on a scenario analysis approach. The model uses a bottom-up vertical modeling approach to assess factors such as energy demand and emissions impacts based on data on pollution emissions, energy conditions, and sectoral activity levels in the mid-section of different scenario environments [15–17].

The model requires the construction of a branching structure based on the availability of data, and from there the data are substituted into the model, which in turn leads to scenario design and future projections from past data [18–20]. The LEAP model was designed to systematically analyze the energy consumption and carbon emissions in Beijing through hierarchical and terminal analysis based on the energy parameters of the city, and is based on three key modules: the terminal demand module, the technology and environment module, and the result output module. The terminal demand module is a synthesis of three parts: terminal demand, conversion and distribution, and resource analysis, which can convert and combine and match distribution points with energy types based on the analysis of energy demand for residential life, provincial and municipal industries, commerce, transportation, and agriculture, while the resource analysis part focuses on the relationship between resource output, storage, and resource import and export [21–23]. The technical and environmental data are mainly based on different open data and terminal technologies to form a technical basis that can be used by the model for in-depth analysis. The resulting output module is the main module used to forecast the energy demand status, resource utilization status, capital and operation related capital consumption, and environmental impact status [24–26]. On this basis, the present study combines the background energy use characteristics and industry characteristics to establish the LEAP (Bj) model, which mainly adopts a tree structure. The specific results are shown in Figure 1.

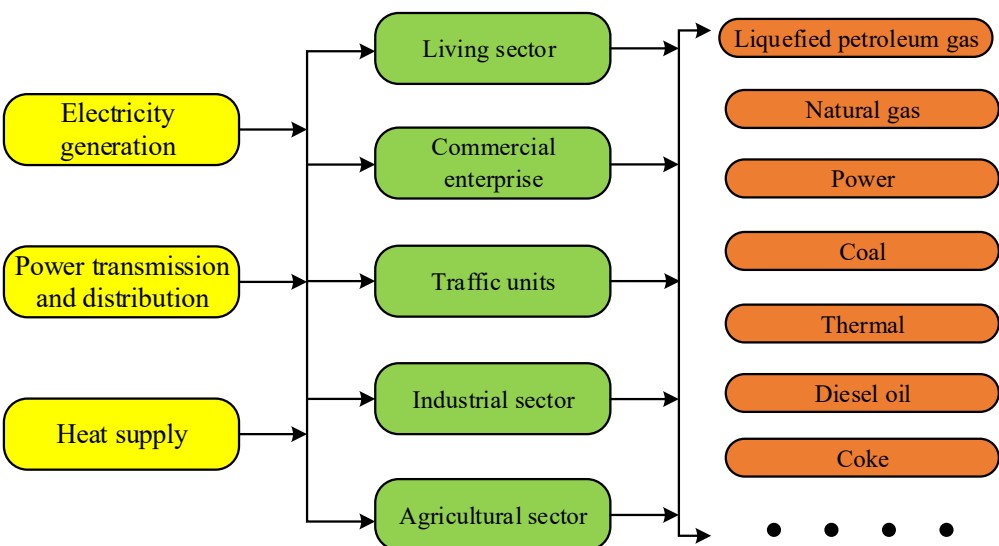

**Figure 1.** LEAP (Bj) model architecture.

Among the main assumptions of the LEAP (Bj) model, the assumptions of the model can be classified into three main types: input variables, control variables, and derived variables. The input variables are the basic data, containing socioeconomic and industry information such as GDP and basic population; the derived variables are indirect variables that are further calculated based on the input variables; and the control variables represent the main variables used in scenario analysis and forecasting, such as energy consumption, pollutant emissions, and other key variables. The equation forLEAP (Bj) Model End-Use Energy Demand Accounting is derived from the interaction of the energy intensity and activity levels, and is calculated as shown in Equations (1)–(3).

$$E = E_i + E_j \tag{1}$$

Among them,
$E$ denotes the total energy demand;
$E_i$ indicates end-use energy consumption;
$E_j$ is the loss of energy formed during the conversion process.

$$E_i = \sum_a \sum_b A_{b,a,i} \times I_{b,a,i} \tag{2}$$

Among them,

$i$ indicates the type of energy source;

$j$ indicates the type of primary energy;

$a$ indicates department;

$b$ indicates the terminal that uses energy;

$A$ indicates the level of departmental activity;

$I$ indicates the energy intensity.

$$E_j = \sum_c \sum_d TR_{d,c} \times \left( \frac{1}{f_{d,c,j}} - 1 \right) \tag{3}$$

Among them,

$TR$ indicates energy conversion products;

$f$ denotes the energy conversion efficiency;

$c$ indicates energy conversion equipment;

$d$ indicates secondary energy.

The formula of the carbon emission calculation tool is as shown in Equations (4)–(6).

$$C = C_1 + C_2 \tag{4}$$

Among them,

$C$ indicates the total carbon emissions;

$C_1$ indicates carbon emissions from end-use energy consumption;

$C_2$ represents the carbon emissions formed during the conversion of energy production.

$$C_1 = \sum_a \sum_b \sum_i A_{i,b,a} \times I_{i,b,a} \times CF_{i,b,a} \tag{5}$$

$CF$ denotes the energy conversion emission factor.

$$C_2 = \sum_j \sum_c \sum_d TR_{d,c} \times \frac{1}{f_{d,c,j}} \times CF_{j,d,c} \tag{6}$$

### 2.2. Selection of Parameters for Each Scenario

In the scenario parameter selection section, this study divides the scenarios into two main categories; one is the baseline scenario, which is a continuation of the scenario formed without any change in the current background energy or the pollution control policies [27]. The others are scenarios formed under the assumption of other conventional possibilities for development, namely, the transportation electrification scenario, the renewable energy development scenario, and the energy efficiency improvement scenario. When establishing the scenario parameters, this study takes the data and change trend in the background statistical yearbook, that is, the recent energy consumption and change trend in Beijing, as the benchmark scenario. On this basis, the study takes the "Fourteenth Five Year Plan" Beijing New Energy Development Plan and the 2035 Long term Target Proposal Document, "Beijing Energy Development Plan during the" 13th Five Year Plan "and other planning policy documents as the possibility basis for different scenarios. The design of parameters under the baseline scenario is shown in Table 1.

The transportation electrification scenario refers to the gradual transition of transportation energy from fuel energy dominance to electrical energy dominance. The scenario of transportation electrification refers to the gradual transformation of transportation energy from fuel energy to electrical energy. Under the development trend of new energy transportation in Beijing, the continuous partial electrification of transportation and ultimate achievement of full electrification of transportation are one of the most likely trends in the development of power added energy in Beijing. The parameters of the transportation sector in Beijing under this scenario are designed as shown in Table 2.

**Table 1.** Baseline scenario parameters.

| Department Classification | Traffic Classification | Variable Content | 2021 Parameter Values | Average Annual Increment |
|---|---|---|---|---|
| Transportation Department | Non-bus passenger transport | Minibus | 5.24 million units | 0.93% |
| | | Average value of minibus mileage | 14,332 km | −0.64% |
| | | Small bus electrification penetration ratio | 5.52% | 5.52% |
| | | Average mileage of medium and large buses | 12,292 km | −0.64% |
| | | Average cab mileage | 62,840 km | −241 |
| | | Percentage of cab electrification penetration | 5.59% | 0.079 |
| | | Number of cabs | 715,117 units | 0.36% |
| | Bus passenger transportation | Number of buses electrified | 28,271 units | −0.20% |
| | | Average electric bus mileage | 55,600 km | 0.20% |
| | | Total annual metro distance | 658,172,300 km | 1.60% |
| | | Traction energy consumption (100 km) | 183.74 kWh/100km | −0.28% |
| | | Annual value of the number of aircraft landings and takeoffs | 649,000 | 0 |
| | | Average takeoff and landing energy consumption | 15.81 tce/time | 0.35 tons tce |
| | Cargo | Number of trucks | 483,700 units | 2.10% |
| | | Average single-year mileage for trucks | 12,600 | 0 |
| | | Diesel truck ratio | 80% | 0 |
| | | Electric truck ratio | 3.67% | −1.75% |
| Post and Telecommunications | Activity Level | Total business volume | 276.1 billion yuan | 29.00% |
| | Energy consumption | Average heat consumption | 0.0073 kwh/yuan | 0.00% |
| | | Electricity consumption average | 0.077 million tce/billion yuan | −22.00% |
| Living Sector | Energy consumption | Electricity for lighting (urban areas) | 8.77 kWm/m$^2$ | 1.40% |
| | | Electricity for lighting (agricultural area) | 8.77 kWm/m$^2$ | 1.40% |
| | | Electricity for electrical appliances (urban areas) | 15.178 billion kWh | / |
| | | Electricity for electrical appliances (agricultural area) | 3471 million kWh | |
| | | Coal consumption (urban areas) | 12.13k tce | −36.00% |
| | | Coal consumption (agricultural areas) | 188,000 tce | −34.00% |
| | | Natural gas consumption (urban areas) | 1,449,500 tce | −0.32% |
| | | Natural gas consumption (agricultural areas) | 307,700 tce | 7.35% |
| | | Heat consumption (urban areas) | 1,682,900 tce | 3.97% |
| | | Heat consumption (agricultural area) | 0.00% | 0.00% |
| | Activity Level | Per capita value of living area (urban area) | 32.55 m$^2$ | 0.00% |
| | | Per capita value of living area (agricultural area) | 47.18 m$^2$ | 0.00% |
| | | Average value of population households (urban areas) | 4.38 people | 0.00% |
| | | Average value of population households (agricultural areas) | 2.78 people | 0.00% |
| Business Sector | Energy consumption | Electricity consumption (per unit area) | 102.19 kwh/m$^2$ | −0.27% |
| | | Total natural gas use | 3.748 million tce | 2.70% |
| | | Liquefied Petroleum Gas | 107,900 tce | −29.00% |
| | | Coal Use | 27,500 tce | −31.00% |
| | | Heat use | 3.018 million tce | 1.70% |
| | | Diesel use | 710,600 tce | 2.28% |
| | Activity Level | Total building area | 386 million m$^2$ | 0.15 billion m$^2$ |

**Table 1.** *Cont.*

| Department Classification | Traffic Classification | Variable Content | 2021 Parameter Values | Average Annual Increment |
|---|---|---|---|---|
| Industrial Sector | Manufacturing Department | Average energy intensity | 0.017 kgce/yuan | −7.51% |
| | | Total annual output | 3527.2 billion yuan | 4.78% |
| | Mining sector | Average energy intensity | 0.0084 kgce/yuan | −8% |
| | | Total annual output | 41.4 billion yuan | −28% |
| | Energy supply sector | Average energy intensity | 0.034 kgce/yuan | −4.30% |
| | | Total annual output | 10,605 billion yuan | 0.068 |
| Agricultural sector | Energy consumption | Coal | 20,300 tce | −44.80% |
| | | Gasoline | 34,000 tce | −1.30% |
| | | Diesel | 31,200 tce | −7.70% |
| | | Power | 217,400 tce | −14% |
| | Activity Level | GDP | 11.46 billion | −6% |
| Heat Sector | Fuel classification | Power | 0.2218 | 0.2218 |
| | | Natural Gas | 0.7782 | 0.7782 |
| Power Sector | Non-renewable Energy | Coal | 780,000 kW | 780,000 kW |
| | | Natural Gas | 9.9 million kW | 19.338 million kW |
| | Renewable Energy | Utilities | 980,000 kW | 980,000 kW |
| | | Wind Energy | 190,000kW | 190,000kW |
| | | Photovoltaic | 530,000 kW | 530,000 kW |
| | Peak utilization rate | Thermal Power | 50.63% | 50.63% |
| | | Wind Power | 16.35% | 16.35% |
| | | Photovoltaic | 10.54% | 10.54% |
| | | Utilities | 11.88% | 11.88% |
| | Transmission loss | Line Loss Rate | 6.73% | 6.73% |

**Table 2.** Scenario parameters of traffic electrification.

| Traffic Classification | Variable Content | 2021 Parameter Values | Average Annual Increment |
|---|---|---|---|
| Non-bus passenger transport | Minibus | 5.25 million units | 0.93% |
| | Average value of minibus mileage | 14,342 km | −1.30% |
| | Small bus electrification penetration ratio | 5.52% | 0.049 |
| | Medium and large buses | 110,400 units | 0.77% |
| | Average mileage of medium and large buses | 12,293 km | −1.40% |
| | Large and medium-sized bus electrification penetration ratio | 5.52% | 0.048 |
| | Average cab mileage | 62,842 km | −243 km |
| | Percentage of cab electrification penetration | 5.59% | 0.079 |
| | Annual value of aircraft takeoff and landing | 6.15 million | 0 |
| | Average energy consumption for aircraft takeoff and landing | 1.5808 tce | 0 |
| Bus passenger transportation | Number of buses electrified | 23,013 units | −0.30% |
| | Average electric bus mileage | 55,800 km | 0.30% |
| | Total metro distance | 658,172,400 km | 1.60% |
| | Power consumption for subway operation | 183.76 kWh/100 km | −0.28% |
| Cargo | Number of trucks | 483,800 units | 2.20% |
| | Average single-year mileage for trucks | 12,700 | 0 |
| | Diesel truck ratio | 80% | −0.075 |
| | Electric truck ratio | 3.69% | 0.075 |

The renewable energy development scenario refers to the increasing utilization of renewable energy under the low carbon transition trend, while the proportion of energy utilization, such as solar energy, in the overall energy utilization is increasing. Thus, the gradual growth of electricity demand in Beijing is met by further utilization of renewable energy. As the second category of solar energy resources in China, Beijing has its own regional characteristics suitable for the development and utilization of solar energy. On the one hand, the use of photovoltaic power generation conforms to the resource transformation trend of reducing carbon and increasing power in Beijing; on the other, it can help Beijing to effectively reduce the proportion of emissions from the power and heat sectors. The design of the conversion sector parameters in Beijing under this scenario is shown in Table 3.

**Table 3.** Renewable energy development scenario parameters.

| Department Classification | Energy Classification | Variable Content | 2021 Parameter Values | 2060 Parameter Values |
|---|---|---|---|---|
| Heat Sector | Renewable Energy | Power | 22.21% | 63.82% |
| | Non-renewable Energy | Natural Gas | 78.79% | 21.31% |
| Power Sector | Fossil Energy | Natural Gas | 9.9 million kW | 0 million kW |
| | | Coal | 780,000 kW | 0 million kW |
| | New Energy | Wind Energy | 200,000 kW | 4.36 million kW |
| | | Biomass Energy | 310,000 kW | 1.87 million kW |
| | | Photovoltaic | 530,000 kW | 41.07 million kW |
| | Power Transmission | Line Loss Rate | 6.76% | 5.46% |

The energy efficiency improvement scenario is a scenario in which Beijing's energy efficiency is improved through energy efficiency retrofits in housing, building retrofits for energy efficiency improvement, and gradual improvement of energy efficiency technologies in the commercial and living sectors, which can meet the city's energy demand while creating a stable control on carbon emissions. Referring to the energy conservation and consumption reduction plan of Beijing during the 13th Five Year Plan period, one of the main directions of Beijing's future development is to improve energy efficiency by reducing the energy consumption of housing, social life, public units, and other innovative technical means. The design of the parameters for each sector in Beijing under this scenario is shown in Table 4.

**Table 4.** Scenario parameters for energy efficiency improvement.

| Department Classification | Variable Content | 2021 Parameter Values | 2060 Parameter Values |
|---|---|---|---|
| Agricultural sector | Electricity consumption intensity index | 1 | 0.7 |
| | Fuel consumption intensity index | 1 | 0.8 |
| Transportation Department | Electricity consumption intensity index | 1 | 0.5 |
| | Fuel consumption intensity index | 1 | 0.45 |
| Industrial Sector | Electricity consumption intensity index | 1 | 0.8 |
| | Fuel consumption intensity index | 1 | 0.7 |
| Business Sector | Electricity consumption intensity index | 1 | 0.75 |
| | Fuel consumption intensity index | 1 | 0.7 |
| Living Sector | Electricity consumption intensity index | 1 | 0.7 |
| | Fuel consumption intensity index | 1 | 0.8 |

### 2.3. Modeling of Parameter Selection under Special Scenarios

For a more comprehensive analysis, this study adds two special scenarios, namely, a partial energy transition scenario and full energy transition scenario, to each individual

scenario. The partial energy transition scenario is a combination of the transportation electrification scenario, renewable energy development scenario, and energy efficiency improvement scenario which is relatively easy to implement. The full energy transition scenario, on the other hand, is a more desirable and difficult to achieve scenario in which electricity substitution in all sectors, including urban agriculture, industry, commerce, and living is increased and the use of non-renewable energy is reduced. In the total energy transition scenario, the consumption of non-renewable energy gradually slows down with changes in policies and the industrial energy use structure, eventually achieving negative growth after reaching peak energy consumption and finally reaching a total non-renewable energy consumption of zero in 2060.

## 3. Result

### 3.1. Analysis of the Results of Energy Demand and Total Carbon Emissions under Different Scenarios

This study analyzes the simulation results of the LEAP (Bj) model for different scenarios on the basis of model construction and scenario parameter selection, and conducts a separate analysis of the comprehensive energy transition scenario. The trends of total energy demand and total carbon emissions in Beijing under different scenarios over time are shown in Figure 2.

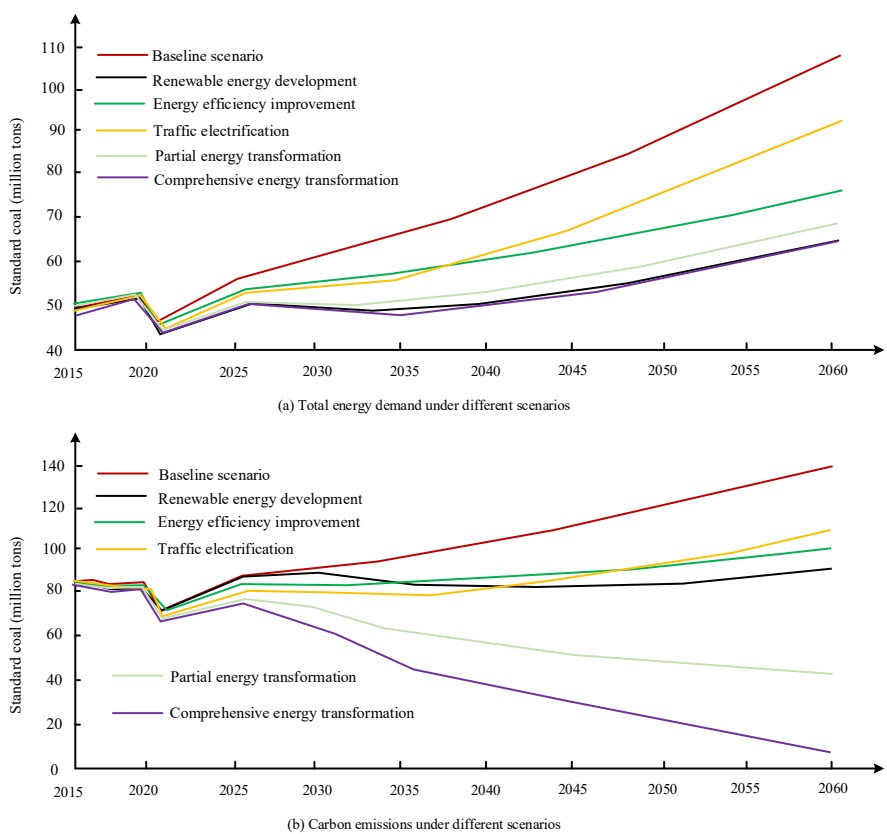

(a) Total energy demand under different scenarios

(b) Carbon emissions under different scenarios

**Figure 2.** Changesintotal energy demand and total carbon emissions.

Figure 2a shows the total energy demand in Beijing under different scenarios, in which the base information is before 2020 and the forecast information is from 2020 to 2060, which can be divided into the baseline scenario, renewable energy development scenario, energy efficiency improvement scenario, transportation electrification scenario, partial energy transition scenario, and full energy transition scenario. It can be seen that the energy demand in the baseline scenario is almost uncontrollable, and shows a rapid growth trend, while the energy demand in the renewable energy development scenario, energy efficiency improvement scenario, and transportation electrification scenario can be controlled, with the overall energy demand under control. The energy demand under

both the partial energy transition scenario and full energy transition scenario is even more controlled, with energy demand of 68.651 million tons of standard coal and 75.759 million tons of standard coal, respectively, of which the energy demand under the full energy transition scenario saves about 2.893 million tons of standard coal. Figure 2b shows the total carbon emissions of Beijing under the different scenarios; it can be seen again that only the partial energy transition scenario and the full energy transition scenario have better control of carbon emissions. Both achieve carbon peak in 2025 with 81.903 million tons and 80.624 million tons, respectively, and achieve reductions to 46.588 million tons and 10.521 million tons in 2060, respectively.

In order to test the degree of influence of different policies on carbon emissions from the energy transition, we conducted a sensitivity coefficient analysis of carbon emissions from energy transition, with the results of the analysis shown in Table 5.

**Table 5.** Carbon emission sensitivity coefficient of energy transformation.

| Policy Type | Policy Implications | Carbon Sensitivity Factor |
|---|---|---|
| Electricity substitution (agriculture) | 1% increase in electricity substitution in agriculture-related non-sectors | 0 |
| Electricity substitution (industrial) | 1% increase in electricity substitution in industry-related non-sectors | −0.17 |
| Electricity substitution (life) | 1% increase in electricity substitution in non-life related sectors | −0.19 |
| Electricity substitution (heat) | 1% increase in electricity substitution in heat-related non-sectors | −0.22 |
| Electricity Substitution (Commercial) | 1% increase in electricity substitution in business-related non-sectors | −0.36 |
| Electricity substitution (transportation) | 1% increase in electricity substitution in transportation-related non-sectors | −0.37 |
| Renewable Energy Development | 1% increase in the proportion of installed renewable energy | −0.43 |
| Energy efficiency improvement | 1% increase in energy use efficiency | −0.86 |

It can be seen that the highest absolute value of the carbon emission sensitivity coefficient for differences in policy types is 0.86 for the energy efficiency promotion policy, followed by 0.43 for the renewable energy development policy, while the absolute value of the sensitivity coefficient of the power substitution type policies in all sectors is below 0.40. Among all sectors, the carbon emission sensitivity coefficients of the commercial sector and transport sector are the most prominent, both reaching above 0.35. It can be seen that the commercial sector and transport sector are the main supporting points in Beijing's power substitution policy, followed by the living sector and the heat sector. It can be seen that the overall sensitivity sequence is energy efficiency improvement, renewable resource development, power substitution in the transportation sector, power substitution in the commercial sector, power substitution in the thermal sector, and finally power substitution in the living sector. Therefore, when formulating energy transformation policy, energy efficiency promotion policy, and renewable energy development policy, these should be taken as the core policy, and other power substitution policies should be taken as the auxiliary policy. When selecting the best power substitution policy, the commercial sector and transportation sector should be taken as the core policy sectors, followed by the thermal sector and the living sector.

*3.2. Analysis of Sub-Sectoral Energy Demand and Carbon Emissions under a Comprehensive Energy Transition*

In this study, when conducting the analysis of energy demand and carbon emissions by sector under the comprehensive energy transition scenario, the analysis is conducted in terms of sectoral energy demand and sectoral carbon emissions, respectively; on this basis,

the carbon emission status under the carbon neutral path is projected. The results of the sectoral energy demand projection are shown in Figure 3.

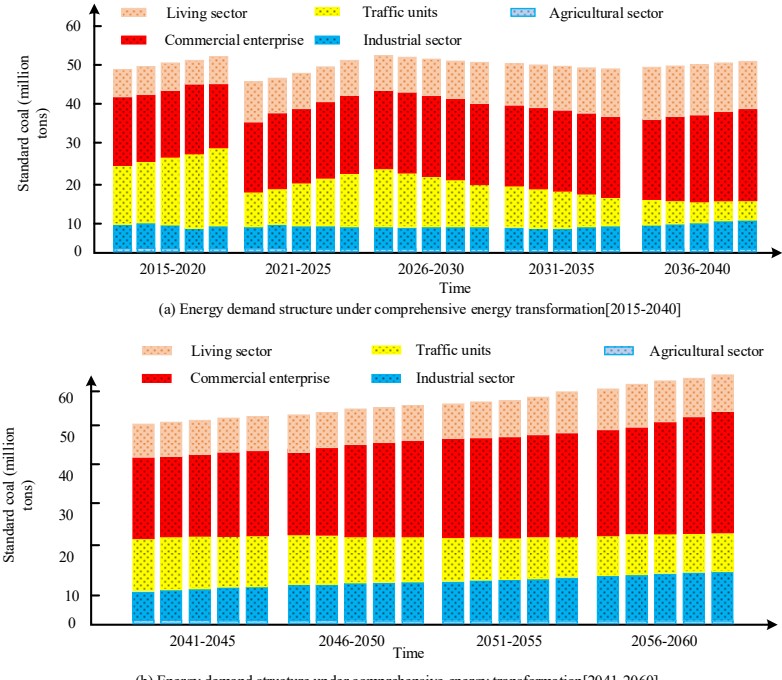

(a) Energy demand structure under comprehensive energy transformation[2015-2040]

(b) Energy demand structure under comprehensive energy transformation[2041-2060]

**Figure 3.** Departmental energy demand forecast results.

It can be seen that, in the full energy transition scenario, the energy demand shows a gradual increase over time, except for the transportation and agriculture sectors, which show a gradual decrease over time, with the agriculture sector showing the least energy demand and the agriculture, industry, and transportation sectors showing more energy demand. The largest decline is in the transportation sector, which is caused by the high electric vehicle share and public rail replacement rate resulting from electrification. The results of the sectoral direct carbon emissions projections for the full energy transition scenario are shown in Figure 4.

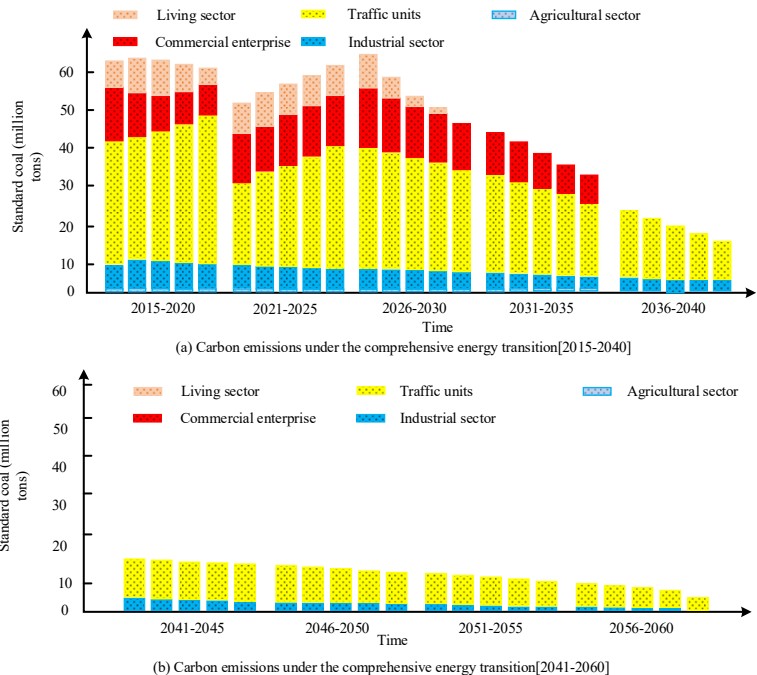

(a) Carbon emissions under the comprehensive energy transition[2015-2040]

(b) Carbon emissions under the comprehensive energy transition[2041-2060]

**Figure 4.** Forecast results of direct carbon emissions of the sector.

It can be seen that in terms of direct carbon emissions, 2035 is a more important time inflection point; before 2035, there are emissions in certain sectors, while after 2035 direct carbon emissions from the agricultural sector, domestic sector, and commercial sector go to zero, and in 2060 direct carbon emissions from the industrial sector show a zero state. At this point, transportation carbon emissions become the only source of carbon emissions. Because electrification technologies such as electric airplanes are in the initial stages of development, it is difficult to achieve zero emissions from transportation in 2060.

Because such a full energy transition is more difficult to achieve, this study conducts a comparative analysis of the carbon neutral pathway; the change in carbon emissions under the carbon neutral energy transition pathway is shown in Figure 5.

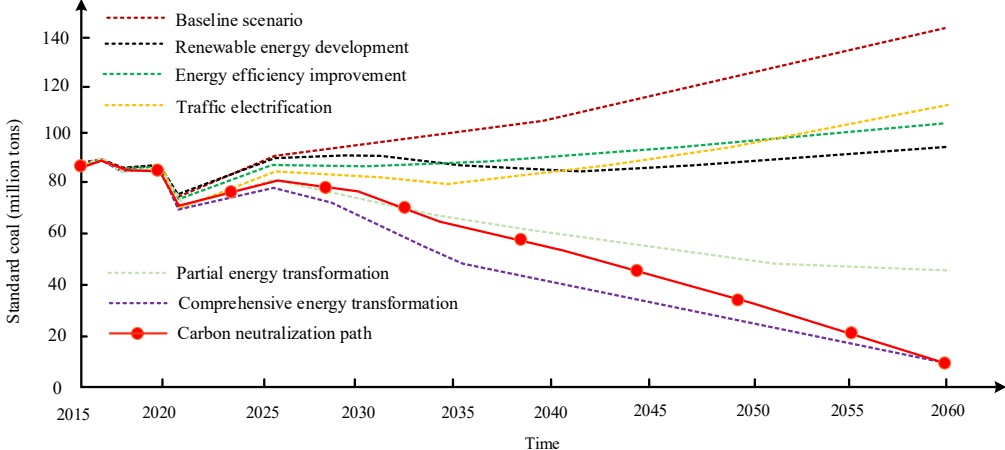

**Figure 5.** Carbon emission changes under carbon neutralization.

It can be seen that the decreasing trend of carbon emissions under the carbon neutral pathway is relatively slow compared to the full energy transition, though it nonetheless shows a rapid and stable decreasing trend overall. The carbon emission folding line reaches a carbon peak in 2025 and begins to decline, with a stable transition to the effect of full energy transition starting in 2035, and finally reaches a carbon emission control effect consistent with full energy transition in 2060. Therefore, a carbon-neutral energy transition path that focuses on the transformation of the power and transportation sectors and is complemented by the commercial, industrial, and domestic sectors is the main path that Beijing should adopt at this time.

## 4. Discuss

With the increasing global environmental problems, carbon emission control and energy transition have become urgent, and the dual carbon target provides a powerful means for implementation of the energy transition. The team of Schlögl R. analyzed the possible energy transition pathways from the perspective of renewable energy sources that can replace fossil fuels. Their study proposed a chemical energy conversion approach to eliminate the environmental pollution caused by fossil fuels and achieve a circular energy economy [26]. The present study explores the possible paths of energy transition in Beijing from the perspective of macro-energy production and consumption. Our the results show that the energy demand is more controlled under the partial energy transition scenario and the full energy transition scenario; the energy demand is 68.651 million tons of standard coal and 75.759 million tons of standard coal, respectively, with the full energy transition scenario saving about 2.893 million tons of standard coal. Safari A. et al. focus on the pre-intermediate phase of the energy transition, and argue that natural gas can be an important support energy source in the pre-intermediate phase of the energy transition when renewable energy is not sufficient to meet energy demand. Therefore, technological upgrading in the area of low-carbon fossil fuels can be an important driver of the energy transition [27]. The results of this study show that 2035 is an important inflection point

in the full energy transition scenario, with direct carbon emissions from the agricultural, domestic, and commercial sectors going to zero after 2035, and the same for the industrial sector in 2060, with transportation emissions becoming the only source of carbon emissions. This is due to the fledgling electrification of aircraft, which makes it difficult to reach a high level of electrification in this sector by 2060. From the perspective of carbon neutrality under the dual carbon target, the team of Li Q. proposed the main research directions of green coal development, intelligent and efficient mining and utilization, and reduced energy consumption and cost through efficient and green coal resource development and utilization technology to achieve the effects of energy conservation and emission reduction [28]. In this study, carbon neutrality is projected as a practical alternative to the full energy transition path, and our results show that although the carbon emission control rate in the early and middle terms is slower than in the full energy transition path, the carbon neutrality path is able to reach the carbon peak in 2025 and gradually transition to the full energy transition effect from 2035 onward, finally reaching the same carbon emission level in 2060. The results show that among the different types of energy transition policy, the sensitivity of energy efficiency improvement policies and renewable energy development policies is high, while the sensitivity of electricity substitution policies in other sectors is relatively low. Thus, in order to achieve a comprehensive energy transition by 2060, Beijing should establish an energy transition system with a set of electricity substitution policies around energy efficiency improvement and renewable energy development policies, and adhere to the carbon neutral path to achieve energy structure optimization and promote energy transition development.

## 5. Policy Suggestions

Our research on the development of Beijing's energy transformation system provides a theoretical direction for the development path of Beijing under its own dual carbon goal planning and an effective source of guidance for the development direction of similar large developing cities under the constraints of environmental and energy conditions. In addition, it provides a feasible path for cities with similar environmental economic development problems under the current international environment. Although this study provides a more specific regional path analysis, Beijing is selected as the main region. Beijing has a certain socio-economic and energy transition foundation, and this analysis may not be fully applicable in other regions with weaker socio-economic and energy transition foundations. How to further weaken the conditionality of regionalization analysis and carry out universality and globalization is amain research direction forthe future.

According to our simulation results, this research proposes the following policies for the development of Beijing's energy transformation.

First, optimization of private car travel policy and promoting the comprehensive development of transportation electrification. The simulation results show that the transport sector is one of the main carbon emission-sensitive sectors in the power replacement strategy, and isthe main sector contributing to emissions reduction. When carrying out policy planning for transportation development, we should simultaneously promote the solidification of electric vehicle development and the elimination and idle policy of old cars, and promote the development of electrification while reducing the use intensity of old cars through a series of policies such as traffic restrictions, subsidies for the exchange of electric vehicles, and carbon taxes. In addition, new energy public transport policies should be implemented while limiting residents' private car use habits.

Second, we should pay equal attention to the development of renewable energy and reform of the power structure. In the process of carbon reduction and power generation, the power demand should be transferred by policy, and the development of renewable energy, such as photovoltaic, should be brought into play to a greater extent in order to carry out new energy power substitution policies. On the one hand, relevant policies can effectively promote the further development of renewable energy technology, while on the other they can increase the consumption capacity and utilization level of new energy,

achieve the integrated development of production and marketing, and make use of policies that indirectly promote the improvement of relevant service markets.

Third, energy efficiency should be improved and the upgrading of power supply and heating infrastructure should be sped up. While developing new energy, the region should focus on the fields of energy conservation and efficiency improvement, achieve a combination of soft and hard guidance through financial support for building energy conservation, public facility energy conservation, energy supply and demand side reform and other aspects, introduce social capital for energy conservation transformation, and fundamentally improve the basis of energy utilization. On this basis, it will be possible to accelerate upgrading of the old power supply and heating infrastructure and encourage the use of heat storage equipment. Finally, we encourage the use of private power equipment and recommend the gradual elimination of traditional energy equipment and promotion of modern electrical equipment in industries such as catering, tourism, and basic industry.

## 6. Conclusions

Based on the background of dual carbon goals, the present study analyzed and predicted scenarios for China's regional implementation of the energy transformation and carbon emission control paths. We studied four main scenarios: a benchmark scenario, traffic electrification scenario, renewable energy development scenario, and energy efficiency improvement scenario. Taking the background as the research object, this study established the LEAP model to analyze the path and effect of Beijing's energy transformation under different hypothetical scenario parameters. Our research results show that under the scenario of comprehensive energy transformation, 2035 is an important turning point. After 2035, the direct carbon emissions of the agricultural sector, living sector, and commercial sector return to zero. At the same time, the carbon neutral path can reach the carbon peak in 2025. In 2035 the transition to comprehensive energy transformation is approached, and finally the control effect of comprehensive energy transformation is achieved in 2060. Relevant departments should optimize the private car travel policy from the policy perspective, promote the comprehensive development of transportation electrification, pay attention to the reform of renewable energy and the electric energy structure, comprehensively improve the energy efficiency level of various industries and basic social life, and promote the upgrading of power supply and heating infrastructure. This study provides a reference for the regional implementation path of global energy transformation, a path proposal for the energy transformation of large and medium-sized cities of the same type, and a theoretical basis for specifying the promotion method of the urban energy transformation and carbon emission reduction path.

**Author Contributions:** K.W. and L.O. conceived the research idea and designed the study. L.O. and Y.W. visited the dairy farms and collected the data. K.W. and Y.W. performed data analysis. All authors discussed the results and wrote the manuscript. All authors have read and agreed to the published version of the manuscript.

**Funding:** This research was supported by the Research Foundation of Education Bureau of Hubei Province, China: The "rural web celebrity" phenomena and application research in new media environment, No. Q20204303.

**Institutional Review Board Statement:** This article does not contain any studies with human participants performed by any of the authors.

**Informed Consent Statement:** Not applicable.

**Data Availability Statement:** The data used to support the findings of this study are available from the corresponding author upon request.

**Conflicts of Interest:** The authors declare no conflict of interests.

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
