# Peer review of "Study of Energy Transition Paths and the Impact of Carbon Emissions under the Dual Carbon Target"

_sustainability, doi:10.3390/su15031967_

Round 1

Reviewer 1 Report

In chapter 4. Discuss, on row 312, the author writes the sentence: The effect of[36] . It seems something incomplete.... In the same chapter, on rows 317/318, there is a similar situation. A part of a sentence appears:. control 317 effect in 2060.......

From the point of view of global impact, the work does not have a very general character. It addresses only one region in Asia, the study cannot be useful in other regions of the world... The last chapter could be completed with a paragraph on how the study will continue with a generalization calculation, for the whole continent, or also for other regions.

Author Response

In chapter 4. Discuss, on row 312, the author writes the sentence: The effect of[36] . It seems something incomplete.... In the same chapter, on rows 317/318, there is a similar situation. A part of a sentence appears:. control 317 effect in 2060.......

Reply: Thank you for your suggestion. It has been updated in rows 316-321 and 324-326 of Chapter IV.

Fr calculation, for the whole continent, or also for other regions.

Reply: Thank you for your suggestion. A new general paragraph has been added in Chapter IV, lines 335 to 340.

Reviewer 2 Report

The article "Study of energy transition paths and the impact of carbon emissions under the dual carbon target" addresses an interesting and important topic. However, the study needs some additions before publication.

The executive summary provides an outline of the background to the study, the stated aim and an outline of the methodology. It is also worth outlining the research gap the article fills and signalling limitations.

Methods presented in a clear and comprehensive manner in relation to the scope of the article.

Presentation and discussion of results presents well. Summaries of results supported by visualisations add value to these considerations.

The conclusions indicate the reference of the research background to the global dimension (line 287-289). The global aspect is very much missing from the introduction. It would have been valuable from this perspective if a literature review (embedding the problem in the global dimension) had been included to move from the global scope to the local level of consideration and research. The title of the article (broad overall) reinforces this claim.

The authors refer to the literature. It is worth ensuring that the introductory aspects are more firmly embedded in the global literature. The 36 items of literature are worth expanding. Literature worth mentioning in terms of strengthening the background: an approach to the problem in Europe with numerous literature and legal references can be found e.g. in: https://doi.org/10.3390/en15155461 and other studies by these authors, other approaches in the article https://doi.org/10.3390/economies10080186, or https://doi.org/10.3390/su11143972.

In conclusion, it is worth highlighting the limitations of the research more and linking the results more to the potential for practical application, thus reinforcing the importance of the article.

The literature record should be done according to journal standards.

Author Response

The article "Study of energy transition paths and the impact of carbon emissions under the dual carbon target" addresses an interesting and important topic. However, the study needs some additions before publication.

The executive summary provides an outline of the background to the study, the stated aim and an outline of the methodology. It is also worth outlining the research gap the article fills and signalling limitations.

Reply: Thank you for your suggestion. Relevant contents have been added in lines 23-27 of the summary.

Methods presented in a clear and comprehensive manner in relation to the scope of the article.

Presentation and discussion of results presents well. Summaries of results supported by visualisations add value to these considerations.

The conclusions indicate the reference of the research background to the global dimension (line 287-289). The global aspect is very much missing from the introduction. It would have been valuable from this perspective if a literature review (embedding the problem in the global dimension) had been included to move from the global scope to the local level of consideration and research. The title of the article (broad overall) reinforces this claim.

Reply: Thank you for your suggestion, and relevant contents have been added in lines 105-123 of the introduction.

The authors refer to the literature. It is worth ensuring that the introductory aspects are more firmly embedded in the global literature. The 36 items of literature are worth expanding. Literature worth mentioning in terms of strengthening the background: an approach to the problem in Europe with numerous literature and legal references can be found e.g. in: https://doi.org/10.3390/en15155461 and other studies by these authors, other approaches in the article https://doi.org/10.3390/economies10080186, or https://doi.org/10.3390/su11143972.

Reply: Thank you for your suggestions. We have expanded the literature focusing on globalization, which is numbered 17-19

In conclusion, it is worth highlighting the limitations of the research more and linking the results more to the potential for practical application, thus reinforcing the importance of the article.

Reply: Thank you for your suggestion. Relevant content has been added at the end of the discussion paragraph, which is located in 357-368 red font.

The literature record should be done according to journal standards.

Reply: Thank you for your suggestion that the document citation format has been standardized again. See the References section.

Reviewer 3 Report

I am grateful to the editor for involving me in discussing the work of this paper and making comments.

In general, I am disappointed with what I consider to be a rather average quality, which can be considered as a simple application of the LEAP model, and the authors' systematic review of the research question is very inadequate, and the novelty of the paper is very limited.

Specifically, there are also the following comments.

(1) Regarding the introduction. The overview is extremely lacking and lacks a systematic organization of current research, especially since the problem studied by the authors is in fact very general and the authors do not discuss the existing contributions to the current research field, which may lead to the possibility that the authors' work in this paper is a simple repetition of the work of their predecessors.

(2) Regarding the novelty. What is the contribution of this paper? I see only a large number of other scholars' work listed, but what is the significance, value, and logic of all these works? What differences does this paper highlight on the basis of these works? Not reflected.

(3) Methodology section. The study area is not even introduced. Why is it valuable to study the authors' work in Beijing? This needs to be articulated in the context of Beijing's economic and social development, which is currently a gap

(4) Realistic basis for scenario setting. A large number of scenarios come from the literature, which is needed but not enough. What is the relationship between these scenarios and the development of Beijing itself? Are these scenarios realistic alternatives for the Beijing region? Or are they just wishful thinking on the part of the authors?

(5) Discussion. The analysis of the results is rather lacking, especially in terms of revelatory discussion, what is the significance of the discussion in this paper? What social problems revealed by the results of this paper will be addressed by future research? What are the limitations of this paper? Not reflected.

I think this paper is more like a detailed report, and the current manuscript does not meet the requirements of an "article".

Author Response

I am grateful to the editor for involving me in discussing the work of this paper and making comments.

In general, I am disappointed with what I consider to be a rather average quality, which can be considered as a simple application of the LEAP model, and the authors' systematic review of the research question is very inadequate, and the novelty of the paper is very limited.

Specifically, there are also the following comments.

(1) Regarding the introduction. The overview is extremely lacking and lacks a systematic organization of current research, especially since the problem studied by the authors is in fact very general and the authors do not discuss the existing contributions to the current research field, which may lead to the possibility that the authors' work in this paper is a simple repetition of the work of their predecessors.

Reply: Thank you for your suggestion. Research contributions have been added in the summary and introduction respectively. See lines 23-17 and 104-123.

(2) Regarding the novelty. What is the contribution of this paper? I see only a large number of other scholars' work listed, but what is the significance, value, and logic of all these works? What differences does this paper highlight on the basis of these works? Not reflected.

Reply: Thank you for your suggestion. The description of research contributions and differences has been added in the summary and introduction respectively. See lines 23-17 and 120-123.

(3) Methodology section. The study area is not even introduced. Why is it valuable to study the authors' work in Beijing? This needs to be articulated in the context of Beijing's economic and social development, which is currently a gap

Reply: Thank you for your suggestion. Relevant content has been added in the first section of the method. See lines 126-134 for details.

(4) Realistic basis for scenario setting. A large number of scenarios come from the literature, which is needed but not enough. What is the relationship between these scenarios and the development of Beijing itself? Are these scenarios realistic alternatives for the Beijing region? Or are they just wishful thinking on the part of the authors?

Reply: Thank you for your suggestions. The reasons for choosing different scenarios have been explained. See lines 211-216, 224-229 and 236-240 for details.

(5) Discussion. The analysis of the results is rather lacking, especially in terms of revelatory discussion, what is the significance of the discussion in this paper? What social problems revealed by the results of this paper will be addressed by future research? What are the limitations of this paper? Not reflected.

Reply: Thank you for your suggestion. Relevant content has been added to the last part of the discussion. See lines 379-390.

I think this paper is more like a detailed report, and the current manuscript does not meet the requirements of an "article".

Reply:Based on the current characteristics of resource transformation and development in Beijing, the study puts forward assumptions for various characteristic scenarios. On this basis, the study establishes a model to predict and analyze the development path for the hypothetical scenarios, and finally proposes a prediction path that deviates from the ideal state, which can provide a theoretical basis for the path planning of energy transformation and development in Beijing, and provide an example for the regional planning of international energy transformation and development.

Reviewer 4 Report

Dear Authors,

Thank you for your work. Below are my observations after reading the article.

The article is a simulation of the possibility of reducing CO2 emissions into the atmosphere through the extensive introduction and promotion of, among others, renewable energy sources.

The Authors strongly focused on communication and the transport industry, but it is the construction sector through production, overproduction (much more aggregates are now produced and extracted than in the 20th century), exploitation and processing that pose a real threat to the environment and humans. The article is optimal, but it does not mention what is the current concentration level of CO2 in the atmosphere, there is no broader information which countries and to what extent and through what sector have contributed to the deterioration of environmental conditions, as a result of which a solution is currently being sought to prevent further environmental destruction within the global sustainable economy, which would give a real message to this publication.

I understand that the authors focused on one region, but the scale of the problem is larger and should be mentioned. In addition, when performing such simulations, attention should be paid to the capabilities of a given country or region, will it be able to handle the given solutions in the current reality?

The prices of fuels and energy in Europe (including the natural gas proposed by the Authors) currently exceed the possibilities of solvency or the use of innovative solutions on the housing market, although in 2021 the analyzes did not provide for such situations. It would be beneficial for the article to present alternative solutions in construction, the possibility of building without harming the environment, without excessive use of natural resources in the form of sand or aggregates.

High investment costs associated with renewable energy? Is the selected region prepared for the proposed transformation as part of improving the ecological situation and reducing the concentration of CO2 in the atmosphere through the use of Renewable Energy Sources? Will the investment be profitable for the potential investor and when will it pay off and to what extent?

What is the base scenario for the Authors in the context of development opportunities and the introduction of e.g. electrification of transport or construction, because it has not been clearly defined? Please specify the pessimistic and optimistic variant, taking into account the investor's possibilities and the payback time of the costs incurred for the indicated investments, not only the reduction of CO2 concentration, as pro-ecological activities.

The publication lacks conclusions, there is only discussion - i.e. conjecture.

In my opinion, the article is more of an overview related to the current situation in one of the largest Chinese provinces. I miss a broader analysis of the situation, a comparison of opportunities also in other regions. Sustainable economy and the associated actions of most countries in the world have defined the path to improving the ecological situation, low-carbon economy and improving the environmental situation by 2050 (which is also based on changes in cement production). According to these principles, rich regions are obliged to help less developed regions, which de facto contribute less to environmental degradation.

I would like to ask the authors to introduce information on the above-mentioned topics.

The article has potential, but in its current version it is a general description of the phenomenon.

Thank you for your work. Regards.

Reviewer

Author Response

The article is a simulation of the possibility of reducing CO2 emissions into the atmosphere through the extensive introduction and promotion of, among others, renewable energy sources.

The Authors strongly focused on communication and the transport industry, but it is the construction sector through production, overproduction (much more aggregates are now produced and extracted than in the 20th century), exploitation and processing that pose a real threat to the environment and humans. The article is optimal, but it does not mention what is the current concentration level of CO2 in the atmosphere, there is no broader information which countries and to what extent and through what sector have contributed to the deterioration of environmental conditions, as a result of which a solution is currently being sought to prevent further environmental destruction within the global sustainable economy, which would give a real message to this publication.

Reply: The study mainly analyzes the energy consumption, pollution sensitivity and policy path of each department in the trend of energy transformation in Beijing. Different scenarios are assumed for the current characteristics of Beijing, and the changes in pollution and energy consumption of each department in the process of energy transformation under different scenarios are analyzed. It can be seen that the pollution sensitivity of the transportation department, the commercial department, the heating department and the living department is relatively high.

I understand that the authors focused on one region, but the scale of the problem is larger and should be mentioned. In addition, when performing such simulations, attention should be paid to the capabilities of a given country or region, will it be able to handle the given solutions in the current reality?

Reply: Thank you for your reading and suggestions. Relevant content has been added to the text. See the last paragraph of the introduction and discussion for details.

The prices of fuels and energy in Europe (including the natural gas proposed by the Authors) currently exceed the possibilities of solvency or the use of innovative solutions on the housing market, although in 2021 the analyzes did not provide for such situations. It would be beneficial for the article to present alternative solutions in construction, the possibility of building without harming the environment, without excessive use of natural resources in the form of sand or aggregates.

Reply: Thank you for your reading and suggestions. This article mainly forecasts and analyzes the possible scenarios and adjustment paths of representative regions in the context of global energy adjustment, and more tends to analyze different departments and policy orientations in the region. It does not carry out a one-way technical analysis on building environmental protection technology. The content of building environmental protection is included in the analysis of living sectors in this article.

High investment costs associated with renewable energy? Is the selected region prepared for the proposed transformation as part of improving the ecological situation and reducing the concentration of CO2 in the atmosphere through the use of Renewable Energy Sources? Will the investment be profitable for the potential investor and when will it pay off and to what extent?

Reply: Thank you for your reading and suggestions. The article mainly conducts hypothesis and quantitative simulation analysis from the policy perspective, and uses direct carbon emissions and energy demand as two main indicators for quantitative analysis. It does not analyze private investment.

What is the base scenario for the Authors in the context of development opportunities and the introduction of e.g. electrification of transport or construction, because it has not been clearly defined? Please specify the pessimistic and optimistic variant, taking into account the investor's possibilities and the payback time of the costs incurred for the indicated investments, not only the reduction of CO2 concentration, as pro-ecological activities.

Reply: Thank you for your reading and suggestions. Basic scenario related content has been added, which is located in line 207-213. The study mainly predicts the change trend of various departments in Beijing under specific assumptions and policy preferences, and does not analyze the investment cost and recovery time of investors.

The publication lacks conclusions, there is only discussion - i.e. conjecture.

Reply: Thank you for your reading and suggestions. The conclusion chapter has been supplemented. See lines 442-463.

In my opinion, the article is more of an overview related to the current situation in one of the largest Chinese provinces. I miss a broader analysis of the situation, a comparison of opportunities also in other regions. Sustainable economy and the associated actions of most countries in the world have defined the path to improving the ecological situation, low-carbon economy and improving the environmental situation by 2050 (which is also based on changes in cement production). According to these principles, rich regions are obliged to help less developed regions, which de facto contribute less to environmental degradation.

Reply: The study not only carried out the book on the current situation, but also carried out the path change prediction and development analysis based on the current situation in the following specific scenarios, and provided the implementation path reference for the global energy transition through regional policy analysis.

I would like to ask the authors to introduce information on the above-mentioned topics.

The article has potential, but in its current version it is a general description of the phenomenon.

Reply: Thank you for your work. Regards.

Reviewer 5 Report

This paper studies potential of energy transition path to achieve dual carbon target. The title mentions "impact of carbon emission", however, the impacts are not emphasized in the manuscript. Besides, this paper is lacking of result validation for the model used. Assumption made is not mentioned in the paper too. Descriptions for Table 1 to Table 5 are too brief, please add more explanation and analysis. Suggestion, especially on the policies will need to be more specific, perhaps a summary on the policies can be added in.

Author Response

This paper studies potential of energy transition path to achieve dual carbon target. The title mentions "impact of carbon emission", however, the impacts are not emphasized in the manuscript. Besides, this paper is lacking of result validation for the model used. Assumption made is not mentioned in the paper too. Descriptions for Table 1 to Table 5 are too brief, please add more explanation and analysis. Suggestion, especially on the policies will need to be more specific, perhaps a summary on the policies can be added in.

Reply: Thank you for your reading and suggestions. The study selects direct carbon emission indicators as the impact analysis indicators, mainly as Figure 4 and related analysis. The paper applies the model to the path prediction, and the main results are reflected in the analysis part. At the same time, the hypothesis of the paper is mainly reflected in the form of the classification of hypothetical scenarios.

In addition, the descriptions in Table 1 to Table 5 have been added, as shown in the red font before and after the table.

Finally, section V, namely policy recommendations, has been added. See lines 389-434.

Round 2

Reviewer 3 Report

This article still has serious problems after modification, especially the introduction, which is completely inadequate to reach the level of scientific papers

First, I think the theoretical discussion in the introduction is very insufficient. At present, it is a simple list of existing studies. I have not found the relationship between these studies and the research proposed by the author. Are all the literature methods listed by the author used in this article? (How is this possible? What is the author's choice? It is not clear at all.) What is the purpose of listing these documents? The author's way of discussion is completely "What Ateam has done, what Bteam has done", which in my opinion is meaningless. It does not support any research questions or views that the author wants to put forward. The author's motivation for research is completely inadequate, and there is no explanation for the shortcomings of existing research. And this way of writing leads to very stiff language. (And L46 is not even a smooth sentence, which makes it difficult for me not to doubt whether the author's attitude is correct)

Second, the author quotes the background of the Ukraine issue (L102-119) in order to heighten its significance. However, the relationship between this and Beijing as the research topic is very weak. Indeed, from the perspective of globalization, China may face some international energy events or political events, but the nature of the issue with China as the theme is completely different from the Ukraine issue. In my opinion, it is a meaningless newsjacking. It does not highlight the characteristic faced by China at all. It is difficult to say that this analysis is practical and scientific. Does the scenario that the author subsequently analyzed involve the war crisis? (I don't think so at all)

In general, the innovation of the article is still limited, but the current analysis results and discussion may in line with scientific standards (honestly, still insufficient). The current quality of the introduction is not enough to make the article accepted at all.

Author Response

This article still has serious problems after modification, especially the introduction, which is completely inadequate to reach the level of scientific papers

First, I think the theoretical discussion in the introduction is very insufficient. At present, it is a simple list of existing studies. I have not found the relationship between these studies and the research proposed by the author. Are all the literature methods listed by the author used in this article? (How is this possible? What is the author's choice? It is not clear at all.) What is the purpose of listing these documents? The author's way of discussion is completely "What Ateam has done, what Bteam has done", which in my opinion is meaningless. It does not support any research questions or views that the author wants to put forward. The author's motivation for research is completely inadequate, and there is no explanation for the shortcomings of existing research. And this way of writing leads to very stiff language. (And L46 is not even a smooth sentence, which makes it difficult for me not to doubt whether the author's attitude is correct)

Reply: Thank you for your suggestions. The cited documents and relevant discussion contents have been reorganized and adjusted, and some documents have been added and deleted. At the same time, the introduction has been rewritten to explore the purpose of other research and compare the ideas of this research.

The modification is as follows:

At present, the research on energy transformation is mainly divided into two aspects, one is to explore from the technical perspective of energy cleaning, the other is to explore from the perspective of social structure and industrial structure. From the perspective of energy cleaning technology, Cheng Z takes the dual-carbon goal as the strategic goal and focuses on the separation and clean and efficient utilization of low-rank coal. Low-rank coal can show strong compatibility and adsorption capacity through effective surface agent improvement, and can provide an auxiliary path for the realization of the dual-carbon goal [7]. In the context of the dual-carbon goal, the Zou C team carried out research from the perspective of sustainable development of mining enterprises, and reduced the economic cost and energy consumption caused by mineral processing and grinding through the ceramic ball replacement technology in the ore grinding machine, so as to achieve the dual improvement of economic and industrial benefits while ensuring the control of grinding fineness [8]. The Wen W team starts with electricity, the only secondary clean energy, and takes the dual-carbon goal as the strategic goal to build the main path of highly clean energy internet. Through the use of the internet and the Internet of Things for real-time adjustment of power supply, it can achieve coordinated control of the power grid, thus laying the foundation for the stable development of the electrified clean energy network [9]. It can be seen that although the development of the detailed technology of energy cleaning is essential, the support of the detailed technology can only play a supporting role at the bottom level in the regional energy transformation. In the regional energy transformation, especially the urban energy transformation, the guidance of policy means is essential, otherwise the area that technology can affect is always limited. The policy is closely related to the energy industry and the energy consumption market. The Idris M N M team took the utilization of bioenergy in line with the characteristics of the local energy structure in Malaysia as an example, explored the pos-sibility of renewable energy sector development policies, analyzed the stability of the energy and bioenergy market under different financial mechanisms, combined the bio-energy policy with the local carbon dioxide emissions, and analyzed the positive role of energy adjustment policy support for carbon emissions [10]. Jiang T analyzed the structural emission reduction between China's power industry and thermal industry from the perspective of energy input and output under the dual-carbon target, taking the energy utilization method and the input-output method as the main decomposition structure of the study. The research results show that the optimized energy intensity, energy compo-sition and energy input mode are conducive to minimizing the carbon dioxide emissions of China's current heating and power sectors, and then use this minimization scheme to ensure the energy conservation and emission reduction effect under the transition phase of the dual-carbon target [11]. This study selected Beijing as the main object. Similar to Jiang T, it also explored the emission reduction of the thermal and power sectors, but not limited to the thermal and power sectors, but included the transportation sector, the industrial sector and the living sector in the analysis and prediction framework. At the same time, with the same idea as the Idris M N M team, the research created a more realizable energy adjustment scenario based on the local natural renewable energy structure characteristics of the background, and analyzed it under the scenario. Only the analysis based on the characteristics of the local energy structure in the study area and the characteristics of local energy consumption is the integrity analysis of the development possibility of the energy market. Based on this point, this study takes both into account.

Second, the author quotes the background of the Ukraine issue (L102-119) in order to heighten its significance. However, the relationship between this and Beijing as the research topic is very weak. Indeed, from the perspective of globalization, China may face some international energy events or political events, but the nature of the issue with China as the theme is completely different from the Ukraine issue. In my opinion, it is a meaningless newsjacking. It does not highlight the characteristic faced by China at all. It is difficult to say that this analysis is practical and scientific. Does the scenario that the author subsequently analyzed involve the war crisis? (I don't think so at all)

Reply: Thank you for your suggestion. The literature on Ukrainian issues has been deleted, and other documents of international significance, such as Argentina, have been supplemented. At the same time, the guiding significance of the literature for this study has been discussed, and the differences between this study and other studies have been described.

The modification is as follows:

On the basis of energy transformation, the research also mainly analyzes the carbon emissions. Carbon emission has always been an important environmental issue accom-panying economic development, and green economy is the inevitable path of global economic development. Li Z Z et al. analyzed the distribution of green investment in 30 provinces in China from 1995 to 2017 from a macro perspective, and tried to find out the role of key macroeconomic factors in reducing carbon emissions. In the process of research, the long-term correlation analysis between variables was realized through co-integration empirical method. The research results show that the effective layout of green investment projects can achieve long-term and short-term carbon emissions, and there is a two-way causal relationship between the two [12]. At the same time, Sun L team studied the impact of China's domestic trade on the transfer of carbon emissions between different provinces. Based on the social network analysis model and the multi-regional input-output model, the study explored the characteristics of carbon emissions transfer in different departments under different provincial environments. The research results show that the eastern region has a more obvious two-way spillover effect, while the western region plays an intermediary role while the central region mainly plays an inflow role [13]. It can be seen that when only discussing carbon emissions and regional space issues, the issue of how to limit carbon emissions can not be implemented and analyzed. This is because carbon emissions are not only a single pollution issue, but also closely related to local energy industry, energy consumption structure, local environment and energy policies. Most of the global carbon emissions come from the consumption of non-renewable energy, so it is necessary to adjust the regional energy structure. In this field, Zheng J and others analyzed the change of carbon dioxide emissions caused by the upgrading of domestic industrial structure and energy efficiency by exploring China's regional economic and energy development and change model under the dual-carbon goal. The research results show that the industrial structure and energy composition are indeed important driving factors for the reduction of carbon emissions at the national level, and the stabilization of carbon emissions can be effectively maintained through interregional cooperation [14]. Yuping L et al. took Argentina as the main research object and analyzed the dynamic image of renewable and non-renewable energy consumption on Argentina's carbon emission level in 1970 and 2018. The research results show that the coordinated development of renewable energy consumption and global economy can effectively reduce carbon emissions [15]. The Sharif A team analyzed the carbon emission change under the combination of renewable energy and non-renewable energy consumption. The research is using the heterogeneous panel causality analysis. The research results show that all variables present a long-term comprehensive nature. At the same time, the degree of degradation of financial development is conducive to less carbon emissions, while the positive development of finance will have a negative impact on carbon emissions [16]. It can be seen that there is a linkage triangle relationship among the three elements of economic development, energy consumption and carbon emissions. To reduce carbon emissions while developing the economy, we need to adjust the energy consumption and production market as a whole, change the old energy path dependence, and improve the energy structure. The research of Sharif A team and Zheng J team confirmed this point. This study is based on the correlation between energy structure adjustment and carbon emissions, taking Beijing as the main research area, taking the development trend of energy economy as the main scenario hypothesis, and analyzing the impact of energy transformation path on carbon emissions by building a long-term energy substitution planning system, so as to achieve regionalization analysis and regionalization development prospect prediction, and provide a theoretical basis for transformation planning for similar cities under the global energy transformation environment.

In general, the innovation of the article is still limited, but the current analysis results and discussion may in line with scientific standards (honestly, still insufficient). The current quality of the introduction is not enough to make the article accepted at all.

Reply:  Thank you for your suggestions. The introduction has been adjusted and rewritten, and the cited documents have been added and deleted. Thank you again.

The modification is as follows:

With the connection of global social demand markets and the deepening of global industrial development and transfer, the problems of climate change and environmental pollution are becoming increasingly serious. The old consumption mode of non-renewable resources will cause excessive consumption of limited resources and se-rious environmental pollution, which is an unsustainable development mode. Therefore, energy transformation, as a strategy that can achieve benign improvement in both envi-ronmental pollution and resources, is currently being constantly tried [1-3]. As an impor-tant development goal and feasible path for energy transformation and energy conser-vation and emission reduction in the future, the dual-carbon target not only needs to achieve breakthrough development in energy production technology, but also needs to achieve a fundamental change in the energy consumption structure, that is, to achieve energy conservation and emission reduction from both energy production and energy consumption. At present, the research on the dual-carbon target has gradually deepened [4-6].

At present, the research on energy transformation is mainly divided into two aspects, one is to explore from the technical perspective of energy cleaning, the other is to explore from the perspective of social structure and industrial structure. From the perspective of energy cleaning technology, Cheng Z takes the dual-carbon goal as the strategic goal and focuses on the separation and clean and efficient utilization of low-rank coal. Low-rank coal can show strong compatibility and adsorption capacity through effective surface agent improvement, and can provide an auxiliary path for the realization of the dual-carbon goal [7]. In the context of the dual-carbon goal, the Zou C team carried out research from the perspective of sustainable development of mining enterprises, and reduced the economic cost and energy consumption caused by mineral processing and grinding through the ceramic ball replacement technology in the ore grinding machine, so as to achieve the dual improvement of economic and industrial benefits while ensuring the control of grinding fineness [8]. The Wen W team starts with electricity, the only secondary clean energy, and takes the dual-carbon goal as the strategic goal to build the main path of highly clean energy internet. Through the use of the internet and the Internet of Things for real-time adjustment of power supply, it can achieve coordinated control of the power grid, thus laying the foundation for the stable development of the electrified clean energy network [9]. It can be seen that although the development of the detailed technology of energy cleaning is essential, the support of the detailed technology can only play a supporting role at the bottom level in the regional energy transformation. In the regional energy transformation, especially the urban energy transformation, the guidance of policy means is essential, otherwise the area that technology can affect is always limited. The policy is closely related to the energy industry and the energy consumption market. The Idris M N M team took the utilization of bioenergy in line with the characteristics of the local energy structure in Malaysia as an example, explored the pos-sibility of renewable energy sector development policies, analyzed the stability of the energy and bioenergy market under different financial mechanisms, combined the bio-energy policy with the local carbon dioxide emissions, and analyzed the positive role of energy adjustment policy support for carbon emissions [10]. Jiang T analyzed the structural emission reduction between China's power industry and thermal industry from the perspective of energy input and output under the dual-carbon target, taking the energy utilization method and the input-output method as the main decomposition structure of the study. The research results show that the optimized energy intensity, energy compo-sition and energy input mode are conducive to minimizing the carbon dioxide emissions of China's current heating and power sectors, and then use this minimization scheme to ensure the energy conservation and emission reduction effect under the transition phase of the dual-carbon target [11]. This study selected Beijing as the main object. Similar to Jiang T, it also explored the emission reduction of the thermal and power sectors, but not limited to the thermal and power sectors, but included the transportation sector, the industrial sector and the living sector in the analysis and prediction framework. At the same time, with the same idea as the Idris M N M team, the research created a more realizable energy adjustment scenario based on the local natural renewable energy structure characteristics of the background, and analyzed it under the scenario. Only the analysis based on the characteristics of the local energy structure in the study area and the characteristics of local energy consumption is the integrity analysis of the development possibility of the energy market. Based on this point, this study takes both into account.

On the basis of energy transformation, the research also mainly analyzes the carbon emissions. Carbon emission has always been an important environmental issue accom-panying economic development, and green economy is the inevitable path of global economic development. Li Z Z et al. analyzed the distribution of green investment in 30 provinces in China from 1995 to 2017 from a macro perspective, and tried to find out the role of key macroeconomic factors in reducing carbon emissions. In the process of research, the long-term correlation analysis between variables was realized through co-integration empirical method. The research results show that the effective layout of green investment projects can achieve long-term and short-term carbon emissions, and there is a two-way causal relationship between the two [12]. At the same time, Sun L team studied the impact of China's domestic trade on the transfer of carbon emissions between different provinces. Based on the social network analysis model and the multi-regional input-output model, the study explored the characteristics of carbon emissions transfer in different departments under different provincial environments. The research results show that the eastern region has a more obvious two-way spillover effect, while the western region plays an intermediary role while the central region mainly plays an inflow role [13]. It can be seen that when only discussing carbon emissions and regional space issues, the issue of how to limit carbon emissions can not be implemented and analyzed. This is because carbon emissions are not only a single pollution issue, but also closely related to local energy industry, energy consumption structure, local environment and energy policies. Most of the global carbon emissions come from the consumption of non-renewable energy, so it is necessary to adjust the regional energy structure. In this field, Zheng J and others analyzed the change of carbon dioxide emissions caused by the upgrading of domestic industrial structure and energy efficiency by exploring China's regional economic and energy development and change model under the dual-carbon goal. The research results show that the industrial structure and energy composition are indeed important driving factors for the reduction of carbon emissions at the national level, and the stabilization of carbon emissions can be effectively maintained through interregional cooperation [14]. Yuping L et al. took Argentina as the main research object and analyzed the dynamic image of renewable and non-renewable energy consumption on Argentina's carbon emission level in 1970 and 2018. The research results show that the coordinated development of renewable energy consumption and global economy can effectively reduce carbon emissions [15]. The Sharif A team analyzed the carbon emission change under the combination of renewable energy and non-renewable energy consumption. The research is using the heterogeneous panel causality analysis. The research results show that all variables present a long-term comprehensive nature. At the same time, the degree of degradation of financial development is conducive to less carbon emissions, while the positive development of finance will have a negative impact on carbon emissions [16]. It can be seen that there is a linkage triangle relationship among the three elements of economic development, energy consumption and carbon emissions. To reduce carbon emissions while developing the economy, we need to adjust the energy consumption and production market as a whole, change the old energy path dependence, and improve the energy structure. The research of Sharif A team and Zheng J team confirmed this point. This study is based on the correlation between energy structure adjustment and carbon emissions, taking Beijing as the main research area, taking the development trend of energy economy as the main scenario hypothesis, and analyzing the impact of energy transformation path on carbon emissions by building a long-term energy substitution planning system, so as to achieve regionalization analysis and regionalization development prospect prediction, and provide a theoretical basis for transformation planning for similar cities under the global energy transformation environment.

Reviewer 4 Report

Dear Authors,

Line 223 and 239 - there should be a space.

I currently have no objections to the substantive side of the article.

Regards

Author Response

Thank you very much for your kind suggestions. The space was added in the revised paper. 

Reviewer 5 Report

Author did the revision accordingly based on the comments.

Author Response

Thank you very much for your kind suggestion "accept".